# Wheat Germ Spermidine and Clove Eugenol in Combination Stimulate Autophagy In Vitro Showing Potential in Supporting the Immune System against Viral Infections

**DOI:** 10.3390/molecules27113425

**Published:** 2022-05-26

**Authors:** Francesca Truzzi, Anne Whittaker, Eros D’Amen, Camilla Tibaldi, Antonella Abate, Maria Chiara Valerii, Enzo Spisni, Giovanni Dinelli

**Affiliations:** 1Department of Agricultural and Food Sciences, Alma Mater Studiorum-University of Bologna, 40127 Bologna, Italy; whittaker.anne@gmail.com (A.W.); eros.damen2@unibo.it (E.D.); camilla.tibaldi2@unibo.it (C.T.); antonella.abate3@unibo.it (A.A.); giovanni.dinelli@unibo.it (G.D.); 2Department of Biological, Geological and Environmental Sciences, Alma Mater Studiorum-University of Bologna, 40127 Bologna, Italy; chiaravalerii@hotmail.it (M.C.V.); enzo.spisni@unibo.it (E.S.)

**Keywords:** SARS-CoV-2, spermidine, eugenol, autophagy, inflammation, human cell lines

## Abstract

Impaired autophagy, responsible for increased inflammation, constitutes a risk factor for the more severe COVID-19 outcomes. Spermidine (SPD) is a known autophagy modulator and supplementation for COVID-19 risk groups (including the elderly) is recommended. However, information on the modulatory effects of eugenol (EUG) is scarce. Therefore, the effects of SPD and EUG, both singularly and in combination, on autophagy were investigated using different cell lines (HBEpiC, SHSY5Y, HUVEC, Caco-2, L929 and U937). SPD (0.3 mM), EUG (0.2 mM) and 0.3 mM SPD + 0.2 mM EUG, significantly increased autophagy using the hallmark measure of LC3-II protein accumulation in the cell lines without cytotoxic effects. Using Caco-2 cells as a model, several crucial autophagy proteins were upregulated at all stages of autophagic flux in response to the treatments. This effect was verified by the activation/differentiation and migration of U937 monocytes in a three-dimensional reconstituted intestinal model (Caco-2, L929 and U937 cells). Comparable benefits of SPD, EUG and SPD + EUG in inducing autophagy were shown by the protection of Caco-2 and L929 cells against lipopolysaccharide-induced inflammation. SPD + EUG is an innovative dual therapy capable of stimulating autophagy and reducing inflammation in vitro and could show promise for COVID-19 risk groups.

## 1. Introduction

Autophagy (meaning “self-eating”) is a highly conserved, eukaryotic self-degradation cellular pathway that serves to ensure homeostasis by removing damaged cytoplasmic cargo, including long-lived proteins and lipids, misfolded proteins and organelles [1,2]. The entire process, termed autophagic flux, is completed when autophagosomes (with sequestered dysfunctional cargo) fuse with lysosomes and the cargo is degraded in autophagolysosomes to generate basic building blocks (free fatty acids and amino acids), which are then released into the cytoplasm for recycling [1,2,3]. 

Autophagy is also involved in the specific disposal of intercellular viral pathogens (virophagy), resulting in the restriction of viral replication and the subsequent containment of infection [4,5,6]. However, several viral pathogens, including the Betacoronavirus genus (β-CoVs) with single-stranded positive sense RNA genomes, “hijack” autophagy machinery to foster replication and avoid degradation [4,7,8]. More specifically, very recent studies, forming the subject of current reviews [4,6,7,8,9,10], have implicated autophagy in severe Acute Respiratory Syndrome Coronavirus 2 (SARS-CoV-2) replication and pathogenesis. 

Given that the World Health Organization (WHO) declared Coronavirus disease 2019 (COVID-19), caused by SARS-CoV-2, a “global pandemic” on 11 March 2020, an urgent appeal was made for research initiatives towards identifying “off-label” or novel drugs [8]. The latter would constitute promising candidates for clinical trials towards the application of approved drugs in the treatment of COVID-19 [11]. Hence, concomitant with vaccination programs, research initiatives are currently centered on investigating the interplay between SARS-CoV-2 and autophagy, with the aim of finding potential autophagy-modulating agents with anti-viral potential for therapeutic interventions [3,6,7,8,10,11].

SARS-CoV-2 has been shown to utilize the angiotensin converting enzyme II (ACE2) for viral entry into the host cells via endocytosis [12,13], involving the direct fusion of the virus with the cell membrane to form endosomes. Released viral RNA is then translated, after which the viral proteins are reported to hijack the unfolded protein response machineries of the endoplasmic reticulum [14] as well as endosomes [8] to generate double-membrane vesicles (DMVs, mimicking autophagosomes) for replication and translation. 

Moreover, SARS-CoV-2 encoded open reading frame (ORF) proteins, ORF3a and ORF7a, were shown to block autophagosomal degradation using distinctive molecular mechanisms, with ORF3a preventing fusion between autophagosomes and lysosomes and ORF7a reducing lysosomal acidity [15,16,17]. Coronaviruses were also reported to exploit deacidified lysosome-mediated exocytosis for the egress of newly synthesized viruses [18]. Reduction of autophagy and the accumulation of lipidified protein 1 light chain 3 (LC3-II, or LC3B) and ubiquitin-binding protein 62 (P62), were also shown to be associated with the SARS-CoV-2-induced decrease of various proteins regulating autophagosome biogenesis at initiation, nucleation, elongation and fusion [11,19].

Currently, literature on SARS-CoV-2, appear to suggest a dual role of promoting viral replication through the early stages of autophagy and blocking later stages to prevent clearance of viral particles and/or antigen presentation [9]. Preclinical investigations suggested repurposing several Food and Drug Administration (FDA)-approved drugs with potential anti-viral effects for therapeutic interventions in clinical trials [20]. 

These drugs, including lysosomotropic inhibitors of viral entry and the endocytic pathway and corticosteroid inhibitors of LC3 recruitment, acted by suppressing autophagy but were associated with severe adverse effects [20] and later discouraged by the FDA [3]. However, given that emerging research indicates that SARS-CoV-2 reduces autophagy [5,11,17,18], evidence so far cautiously points towards an enhanced clearing of SARS-CoV-2 infection with agents that induce autophagy [9].

SARS-CoV-2 infection was shown to facilitate the degradation of Beclin-1, a key regulator of autophagy, thereby, decreasing autophagy [11]. Of great interest, the exogenous administration of autophagy-inducing spermidine (SPD) resulted in increased autophagy and the inhibition SARS-CoV-2 propagation by 85% [11]. More recently, SPD was also shown induce autophagy by both direct inhibition of the rapamycin complex 1 (mTORC1) and upregulation of SQSTM1/p62 expression [21]. 

Moreover, SPD, an endogenous polyamine metabolite, induced the synthesis of the autophagy transcription factor EB (TFEB), a key regulator of proteins responsible for autophagy and mitochondrial respiration [22,23]. Given that SPD and TFEB are reduced in T cells with advancing age [23,24], SPD supplementation effects on autophagy are recommended to improve lifespan [25] and vaccine immunogenicity [22], as well as in reducing age-associated pathologies in the elderly [24,26,27].

Given the reputed health benefits of SPD already present in daily human nutrition, well-designed in vitro and in vivo studies were suggested to clarify putative benefits of polyamine supplementation before or during SARS-CoV-2 infection [11]. To this end, the present study, based on the results of Gassen et al. [11], aimed to investigate the effect of wheat germ SPD extract, in a unique combination with eugenol (EUG) on autophagy. 

This incentive formed part of one of the European Institute for Innovation & Technology (EIT) FOOD (www.eitfood.eu, accessed on 20 July 2021) projects using natural products that can support either the prevention of COVID-19 or in the treatment of individuals with a higher risk for more severe outcomes of COVID-19. 

Although COVID-19 severity mostly ranged from asymptomatic to mild symptoms (such as fever, cough, shortness of breath, sore throat, muscle ache and gastrointestinal symptoms), in some individuals, severe disease progression has been linked to the so-called “cytokine storms”, initiated through rapid virus propagation and uncontrolled inflammatory response. These individuals include the elderly and those with well-known comorbidities related to systemic inflammation, such as obesity, atherosclerosis, type 2 diabetes, hypertension and asthma [28]. EUG has long been of interest as an anti-viral compound, effective at inhibiting the replication process of a numerous viruses [29,30] and of recent potential interest in the treatment of SARS-CoV-2 [31].

The basis for selecting a dual therapy was the potential advantage of using two components with complementary attributes such as anti-viral and anti-inflammatory activities [28]. SPD was selected for reported benefits of increasing autophagy and inhibiting SARS-CoV-2 propagation [11]. Instead, EUG was selected for known anti-viral activity [29,30,31]. In addition, there is a particular interest in our research group for the anti-oxidant and anti-inflammatory properties of EUG-rich essential oils [32]. 

Interestingly, essential oil single molecules, such as EUG, are natural and inexpensive and as such are lacking sponsors for trials needed to validate their therapeutic efficacy [32]. However, a low production cost would render such a potential component more accessible to the public and for this reason EUG was selected for investigation in combination with SPD. Unlike SPD, which is reported to stimulate autophagy, from a single paper using cell lines, EUG was only proposed as an inhibitor of autophagy [33]. Therefore, it was important to test whether EUG did not produce an antagonistic effect with SPD on autophagy. 

Hence, the objective of the present study was to investigate the autophagy efficacy of SPD and the potential effect on autophagy exerted by EUG, both alone and in combination. To this end a series of human cell lines were used, including those from lung, brain and intestinal origins, organs most affected by COVID-19. The SPD and EUG were derived from natural plant sources, as the objective was to simulate natural nutritional intake of the active components. In order to be administered in small doses, the active ingredients were used on cells in a concentrated form (SPD and pure EUG).

## 2. Results

### 2.1. Establishing the Optimal Spermidine and Eugenol Concentrations Using L929, U937 and Caco-2 Cell Lines

To identify the optimal concentration of wheat-germ extracted SPD without toxic effects, the 3-(4,5-Dimethyl-2-thiazolyl)-2,5-diphenyl-2H-tetrazolium bromide (MTT) assay was selected to evaluate cytotoxicity in L929 fibroblasts, intestinal Caco-2 cells and U937 monocytes [34]. A series of SPD concentrations (0.3–0.9 mM) suspended in Dulbecco’s Modified Eagle’s Medium (DMEM) for L929 and Caco-2 and Roswell Park Memorial Institute Medium (RPMI) for U937 were incubated for 24 h. Cell proliferation was then compared to both an untreated control as well as a starvation control to induce autophagy. 

Relative to the untreated control, there was no significant reduction in cell proliferation in the starvation condition or the 0.3 mM SPD treatment in the L929, U937 and Caco-2 cell lines expressed as an average response (Figure 1A). In contrast, at concentrations ranging from 3 to 9 mM, SPD was shown to significantly reduce proliferation (Figure 1A). Cytotoxicity was then, similarly, evaluated in relation to varying EUG contents both alone and in combination with SPD in the three cell lines (Figure 1B). 

With respect to the untreated control, there was no cytotoxic effect on cell proliferation in response to either the starvation control, 0.2 mM EUG alone, or the combined treatment (0.3 mM SPD and 0.2 mM EUG) on the three cell lines expressed as an average (Figure 1B). There was a slight negative effect with the 0.3 mM SPD (Figure 1B), however no effect on cell viability was evident (Figure 1C). Therefore, the concentrations of 0.3 mM SPD and 0.2 mM EUG alone as well as 0.3 mM SPD and 0.2 mM EUG in combination were selected for all subsequent analyses.

Cell vitality (evaluated with blue trypan exclusion assay) was measured and the average response of the L929, Caco-2 and U937 lines was recorded after a 24 h exposure to 0.3 mM SPD and 0.2 mM EUG alone and in combination. Cell vitality was comparable to both the untreated and starvation controls (Figure 1C). Analyzing the cell lines singularly, no differences were detectable indicating that the SPD, EUG and SPD + EUG treatments had no negative impact on the L929, Caco-2 and U937 cell viability (Figure 1D).

### 2.2. Effect of Spermidine and Eugenol on LC3-II Marker Detection in L929, U937, SHSY5Y, HUVEC and HBEpiC Cell Lines

The efficacy of SPD, EUG and the combined SPD-EUG treatments in stimulating autophagy were then investigated using L929 cell lines. LC3-II, produced from the covalent linkage of cytosolic LC3-I to the phosphatidylethanolamine (PE) lipid on the surface of the phagophore, is widely accepted as the gold standard or hallmark for autophagosome formation (increased autophagy) and was thus employed as a marker in the present study. 

The protocol for LC3-II marker staining using immunocytochemistry/immunofluorescence green (FG) was tested using the untreated control and starvation autophagy-inducer control on L929 cell lines. Clear punctate staining of LC3-II was evident in the L929-starved cell lines after 4 h (Figure 2A). Then using a 4 h exposure time to SP, EUG and SP + EUG significantly increased LC3-II marker staining (with both FG and red chromogen) as was shown to be evident in the L929 cells compared to the untreated control (Figure 2B,C). 

From the red chromogen images, the percentage of LC3-II stained cells in response to the treatments, was calculated after both 4 h and 24 h relative to the control (Figure 2D). In the L929 cells, there was a significant and comparable increase in the percentage of LC3-II marker-stained cells between all three treatments after 4 h, followed by a significant decline after 24 h (Figure 2D).

Viability of the L929 cells was evident from Figure 1D but was repeated using the MTT assay in response to the selected treatments after 24 h (Figure 2E) as well as 48 h (Figure 2F) to verify that the selected concentrations of SPD and EUG did not induce autophagy-mediated apoptosis. No negative effect was evident on the L929 cells in response to the treatments (Figure 2F).

Confirmation of LC3-I (16–18 kDa) conversion to LC3-II (14–16 kDa) as an indicator of increased autophagy was verified by Western Blotting (Figure 2G). The pixels were quantified with respect to the control (Figure 2H) and increased levels of LC3-II were evident in all the treatments.

The treatments U937 cells in suspension showed a significant albeit negligible increase in LC3-II marker-stained cells with respect to the untreated control that was comparable between the three treatments. Under starvation, there was a marginal but significant increase in autophagy (Figure 2H).

Given that SARS-CoV-2 is transmitted through airway epithelial cells as the first gateway for viral invasion [35] and that neurological complications [36], as well as thrombotic and microvascular complications [37], are also common in patients affected by SARS-CoV-2, a human bronchial epithelial cell (HBEpiC) line, a human neuronal cell line (SHSY5Y) and a human umbilical vein endothelial cell (HUVEC) line were investigated in the present study. Moreover, given previous studies have shown that SPD plays an important role in stimulating autophagy in bronchial [11] and neuronal cells [27], the objective was to compare the effect of SPD and EUG alone and in combination on stimulating autophagy in these cell types.

SPD (0.3 mM) and EUG (0.2 mM), alone and in combination, were similarly suspended in the appropriate cell mediums, respectively, for a period of 4 h. Following the staining of LC3-II marker with fast red chromogen, negligible staining in the untreated control cells of SHSY5Y, HBEpiC and HUVEC was evident after the 4 h exposure period compared to the SPD, EUG and SPD + EUG treatments, respectively (Figure 3A,C,E). 

There was a significantly higher percentage of LC3-II marker stained SHSY5Y and HUVEC cells after exposure to both EUG and SPD + EUG compared to SPD alone (Figure 3B,D), with all treatments inducing a significantly higher LC3-II marker staining than in the untreated control. Instead, there was a significantly higher percentage of LC3-II marker stained HBEpiC cells after exposure to both SPD and SPD + EUG compared to EUG alone (Figure 3F). Of interest, a lower percentage of LC3-II marker-stained cells were evident for HUVEC (ca 6–9%) compared to the SHSY5Y (ca 17–25%) and HBEpiC cells (ca 22–37%) (Figure 3B,D,E).

From the MTT assay, there were no toxic effects from the SPD and EUG treatments on the SHSY5Y, HUVEC and HBEpiC lines, all showing a comparable percentage to the control after 24 h (Figure 3G–I). There was a significant effect for the SPD + EUG treatment on all the cell lines, ranging from approximately 75–90% of the control. Cell viability in the starvation control was significantly lower than that for all treatments (Figure 3G,H,I).

### 2.3. Effect of Spermidine and Eugenol on LC3-II Marker Detection and 20 Human Autophagy Array Proteins in Caco-2 Cells

Gastrointestinal symptoms are prevalent in patients with SARS-CoV-2. The Caco-2 cell line, primarily used as a model of the intestinal epithelial barrier, was used as the model system to investigate autophagy in the present study. This cell line is also one of the most used lines in the study of SARS-CoV-2 [38]. Clear punctate staining of LC3-II with FG was evident in the Caco-2-starved cell lines, as well as in cells exposed to SP, EUG and SP + EUG, after 4 h (Figure 4A,B). 

Similarly, using red chromogen (Figure 4C), increased LC3-II marker staining was shown to be evident in the cells exposed to treatments compared to the untreated control. The percentage of LC3-II stained cells from the red chromogen images attributable all treatments, after both 4 and 24 h, was significantly higher than the control (Figure 4D). More specifically, after both 4 h (Figure 4E) and 24 h (Figure 4F), LC3-II staining after exposure to the SPD + EUG treatment was significantly higher than that after to exposure to SPD and EUG alone.

The viability of the cells using the MTT assay was examined after both 24 h (Figure 4G) as well as 48 h (Figure 4H), given that apoptosis has been reported in cancer cell lines after exposure to EUG. The EUG impacted on the viability of the Caco-2 cell lines after 48 h (Figure 4H), but the effect can be considered minimal given that viability was approximately 75% with respect to the untreated control.

LC3-I (LC3A) lipidation to LC3-II (LC3B), which is used as the hallmark measure of autophagy, forms but one stage within a continuous flux. Given that autophagy is a multi-step pathway involving different protein complexes, it has been recommended that flux be analyzed using more than one method [9]. To this end, to evaluate flux encompassing initiation of DMVs, nucleation to form phagophores, elongation of phagophores to form autophagosomes and finally the fusion of autophagosomes to the lysosomes [1,2], the proteomic profile was analyzed with the Human Autophagy Array. The array permitted the semi-quantitative detection of 20 human proteins (positioning indicated by Figure 5C) extracted from Caco-2 cells after a 4 h exposure to the SPD, EUG and SPD + EUG treatments as compared to the control (Figure 5A).

LC3A and LC3B were expressed in the Caco-2 cells exposed to SPD, EUG and SPD + EUG (Figure 5A), thereby, corroborating results in Figure 4D. No upregulation was evident in the control. The only proteins expressed in the control Caco-2 cells were P62 and Nijmegen breakage syndrome 1 (NBS1) that were not significantly different from the treatments (Figure 5A,B). 

Although P62 is also commonly used as a marker of autophagy (through direct binding with LC3B in the recruitment of autophagy cargoes), in cancer cells p62 expression is elevated [39], as is NBS1 [40]. Likely the expression evident in the control Caco-2 cells reflected the signaling (oncogene) activities of these proteins, which were not autophagy-associated [39,40]. The remaining autophagy-related proteins were not expressed in the control (Figure 5A,B) showing that autophagy was not upregulated.

The expression of all the remaining proteins, associated with either the initiation, nucleation, elongation, or autophagosome-lysosome fusion stages of autophagy were all upregulated in the Caco-2 cells exposed to SPD, EUG and SPD + EUG, respectively, compared to the control (Figure 5A,B). 

The cohort of autophagy-related (ATG) proteins regulating conventional autophagy, ATG3, ATG5, ATG7 and ATG10 (essential core proteins required for LC3A lipidation) [41,42] were more significantly expressed in the EUG treatments compared to the SPD treatment alone (Figure 5A,B). ATG13 involved in both the initiation stage (induction of the ULK1 complex) as well as the autophagosome–lysosome fusion stage [43], was equally expressed in all three treatments (Figure 5A,B). Similarly, Beclin-1, important in both the nucleation and autophagosome–lysosome fusion stage of autophagy [9,11] was also expressed equally between all treatments (Figure 5A,B), as was the lysosomal associated membrane protein 1 (LAMP1), which is used as an autophagic lysosome marker (Figure 5A,B). The same was evident for the ATG4 proteins (responsible for LC3A formation) (Figure 5A,B).

### 2.4. Effect of Spermidine and Eugenol on LC3-II and CD14 Marker Detection in Reconstituted Intestinal Equivalents Using Caco-2, U937 and L929 Cells

Three-dimensional (3D) co-culture systems provide more physiologically and structurally relevant in vitro models for studying in vivo cellular responses [44]. In the present study, the intestinal mucosa was simulated by seeding Caco-2 cells (representing the epithelial monolayer) above U937 monocytes and L929 fibroblasts embedded in a collagen layer (representing the extracellular matrix [ECM]-rich lamina). 

Hematoxylin and eosin (H&E) staining of the 3D intestinal equivalents showed the intact preservation of the Caco-2 cells (Figure 6A). LC3-II red chromogen staining was evident in the Caco-2 cells in response to the SPD, EUG and SPD + EUG treatments (Figure 6A). Quantification of the LC3-II staining in pixels was shown to be comparable between the three treatments and significantly higher than that in the control (Figure 6B). Chromogen-red LC3-II staining of cells (U937/L929) in the ECM was also evident (Figure 6A).

Given that autophagy has been shown to play a decisive role in the differentiation of monocytes into macrophages [45,46], verification of autophagy was measured in the form of U937 monocyte activation and subsequent differentiation into macrophages, demonstrated by specific staining of cluster of differentiation 14 (CD14), a glycolipid-anchored membrane glycoprotein expressed on both monocytes and macrophages, enabling the identification of these cells [47]. In Figure 6A, fast-red coupled CD14 staining was evident in response to the treatments. 

Moreover, the migration of the activated monocytes towards the Caco-2 layer was clear in response to the EUG and the SPD + EUG treatments (Figure 6A, demonstrated by the arrows). Quantification of the CD14 stained pixels (Figure 6C), showed a significant increase in the SPD, EUG and SPD + EUG treatments in comparison to the control.

### 2.5. Effect of Spermidine and Eugenol Pretreatment on LC3-II Expression and Inflammation in Caco-2 and L929 Cells Treated with the Lipopolysaccharide

SARS-CoV-2 is well reported to pose a greater risk to people with pre-existing conditions that amplify inflammation. Thus far, the present study has shown SPD, EUG and SPD + EUG to be effective in stimulating autophagy. However, an important requisite for the efficacy of the SPD and EUG treatments would necessitate the stimulation of autophagy to protect the cells against excessive inflammation. 

To test this aspect, Caco-2 cells and L929 cells were pretreated with SPD, EUG and SPD + EUG for 1 h before being subjected to lipopolysaccharide (LPS), a known pro-inflammatory elicitor, for 24 h. A lower content of both SPD (0.1 mM) and EUG (0.05 mM) was shown to be as effective in inducing autophagy and were, therefore, used in this experiment. In both Caco-2 cells (Figure 7A,B) and L929 cells (Figure 7D,E), stimulation of autophagy in response to the treatments was evident from the significantly increased presence of LC3-II protein in Western blots (Figure 7A,D), compared to the respective untreated controls and LPS treatments alone.

The increase in autophagy on inflammatory effects impacting on cell proliferation was then tested. LPS alone caused a significant reduction in the proliferation (MTT assay) of both Caco-2 (Figure 7C) and L929 cells (Figure 7F). Interestingly, the treatments augmented cell viability, despite the presence of LPS. Hence, inflammation-related effects were significantly lessened, with viability largely comparable to the untreated control in the presence of the treatments (Figure 7C,F).

## 3. Discussion

Age-related characteristics, including loss of proteostasis, mitochondrial dysfunction, deregulated nutrient signaling and genomic instability, all constitute “hallmarks of aging” that involve impaired autophagy [48]. In turn, impaired autophagy is responsible for a defective immune response against viral infections, chronic inflammation and reduced vaccine immunogenicity, collectively constituting “hallmarks” of SARS-CoV-2 infection risk factors in the elderly [23,49,50]. Interestingly, decreased autophagy/mitochondrial functioning in both aged- and SARS-CoV-2-infected cells share a common denominator, that being a decreased SPD content [11,22,23,24,25,26,27]. 

Using a novel procedure to extract SPD from a wheat germ, the present study not only substantiated the efficacy of SPD in stimulating autophagic flux in various cell lines (derived from tissue sources susceptible to SARS-CoV-2 infection), but interestingly highlighted a comparable efficacy of EUG in stimulating autophagy. Given that the rapid development of effective treatments to control SARS-CoV-2 is a worldwide priority, the advantage of dual therapies with both antiviral and anti-inflammatory activity was highlighted [28]. To this end, the present study was the first to investigate the autophagy stimulating efficacy of SPD in combination with EUG.

A concentration of 0.3 mM SPD was shown to be effective in stimulating autophagy (using the hallmark measure of increased LC3-II) in six cell lines (L939, U937, HUVEC and SHSY5Y, HBEpiC and Caco-2). Of the six cell lines, autophagy was stimulated to a greater extent in the intestinal Caco-2 and bronchial HBEpiC cell lines. These results corroborated previous research showing that the exogenous application of 0.33 mM SPD to primary lung cells and intestinal organoids resulted in autophagy induction, which in turn was shown to limit SARS-CoV-2 propagation by 85% [11]. Moreover, using the Caco-2 cells as a model, SPD exposure alone stimulated a range of human autophagy proteins, evidencing increased flux over multiple steps (initiation of DMVs, nucleation to form phagophores, elongation of phagophores to form autophagosomes and finally the fusion of autophagosomes to the lysosomes) [1,2].

A concentration of 0.2 mM EUG selected for the well-documented antiviral, anti-inflammatory and antithrombotic effects [31], was also shown to stimulate autophagy (LC3-II expression) in the six cell lines. Additionally, using the Caco-2 cells as a model, EUG exposure alone, similarly, stimulated the same set of human autophagy proteins noted for SPD. Interestingly, in a recent review evaluating the potential of natural products in treating COVID-19 specifically via autophagy, EUG was listed as an inhibitor of autophagy [51], citing a single paper by Dai et al. (2013) [33]. The concentration used by the latter authors to inhibit autophagy was 7-fold lower than that shown to stimulate autophagy in the present study [33]. A stimulatory protective effect of 0.1 mM EUG in vitro on HT22 cells was also recently demonstrated [52], where EUG was shown to specifically induce autophagy at the initiation stage via inhibition of mTORC1 and AMPK.

In the present study, 0.2 mM EUG alone was shown to result in a higher semi-quantitative expression ATG5, ATG7 and ATG10 in Caco-2 cells than that noted for 0.3 mM SPD alone after a 4 h exposure. Comparable effects were noted between EUG and SPD alone for ATG3, ATG4A, ATG4B, ATG13 and Beclin-1. Given numerous recent reports on SPD stimulation of autophagy [11,21,22,23], the effect of EUG can likewise be considered significant, based on the increased expression of several crucial core proteins at all stages of autophagic flux: initiation (ATG13) [43]; nucleation (Beclin-1) [11]; autophagasome elongation (ATG3, ATG5 and ATG7) [41,42] and autophagosome-lysosome fusion (Beclin-1; ATG13) [11,53]. Interestingly, increased ATG3, responsible for linking LC3-I to PE, with the help of the ATG12–ATG5 conjugate [42], has also been shown to been inversely associated with apoptopic-inducing caspase-8 expression, placing ATG3 as a novel link between apoptosis and autophagy [54].

Most of the research on EUG has been focused almost exclusively on investigating the effect of this compound in stimulating apoptosis in cancer cell lines, in which a role for autophagy-mediated apoptosis has also been reported [55,56,57,58]. In the present study, autophagy in Caco-2 cell lines was stimulated to comparable levels after both a 4 h and 24 h exposure to EUG, an effect that did not converge on the apoptosis pathway within that timeframe. 

Moreover, the significant expression of ATG3 (antagonist of caspase-8) in Caco-2 cells after a 4 h period potentially suggest that the early apoptosis events were not activated in response to EUG [54]. Nonetheless, after 48 h, cell viability (MTT assay) in response to the EUG treatments was comparable to the starvation control, evidencing pro-oxidant activity, albeit not extensive. The timeframe generally reported to measure apoptosis in cancer cell-lines in response to EUG is 24 to 48 h [56,58,59,60]. 

Given the that pro-oxidant effects of EUG have been shown to be both concentration dependent as well as cell-type dependent [56,58,59,60], it was important to test a non-cancerous cell line to exclude pro-oxidant side effects after 48 h. The non-cancerous fibroblast (L929) cell lines were unaffected by EUG after 48 h. In fact, the concentration of 0.2 mM EUG used in the present study to stimulate autophagy (thereafter also 0.05 mM EUG) was below the threshold of 0.38 mM (62.1 μg/mL), above which tissue-damaging pro-oxidant effects are usually observed [56,58,60].

Of great relevance, the present investigation is the first to report the efficacy of the exogenous administration of 0.3 mM SPD and 0.2 mM EUG as a dual therapy to enhance autophagy. Previously, both SPD and EUG were singularly promoted as natural products for use as adjuvant therapeutics for the management of the COVID-19 pandemic to prevent SARS-CoV-2 infection and replication [11,31,51], with only SPD showing potential for stimulating autophagy [11]. 

Based on LC3-II detection, SPD + EUG produced neither a synergistic stimulating effect nor an antagonistic effect on autophagy in the L929, U937, HUVEC, SHSY5Y and HBEpiC cells when compared to the administration of the substrates individually. Only in the Caco-2 cells did SPD + EUG produce a significantly higher LC3-II expression than that of both SPD and EUG alone. Nonetheless, the SPD + EUG treatment stimulated a range of human autophagy proteins, evidencing increased autophagic flux over multiple steps. As with the administration of SPD and EUG alone, the combination treatment also stimulated complete autophagic flux, corroborated from activated U937 differentiation into macrophages. 

Prior to macrophage transformation, monocyte differentiation necessitates the stimulation stage, coinciding with the initiation of autophagosome development, as well the adherence stage, coinciding with degradation of cargo in the autophagolysosomes [46]. In the 3D model, mimicking the intestinal mucosa [44], the migration of CD14 differentiated monocytes/macrophages in the ECM towards the Caco-2 cells were demonstrated. In such an instance, monocytes and their macrophage progeny (critical for innate immunity) would potentially then serve in immune-related functions, such as phagocytosis of invading pathogens and in the homeostasis and repair of damaged tissue [46,61].

Given that defective autophagy is responsible for chronic inflammation [23,49,50], and chronic inflammation is a well-known risk factor for more severe outcomes of COVID-19, the efficacy of the dual treatment would depend not only on stimulating autophagy but also reducing inflammation [28]. COVID-19 and the risk of SARS-CoV-2 poses a particular risk to people living with preexisting conditions that impair immune response or amplify pro-inflammatory responses [62]. 

Using Caco-2 and L929 cells, the comparable benefits of both SPD and EUG alone, and in combination, as pre-treatments to increase autophagy and thereby reduce inflammation were evident. Although the underlying mechanisms were not elucidated, the present study corroborated previous reports showing that by stimulating autophagy, cells were protected against LPS-induced inflammation and resultant pro-oxidant effects [63,64,65]. 

As such, a food-based supplement containing the innovative combination of both wheat germ SPD and clove EUG demonstrate great potential either as a SARS-CoV-2 preventative treatment. This is particularly relevant both for the elderly and individuals that are at a greater risk for more severe outcomes of COVID-19. By stimulating autophagy, the contribution of both SPD and EUG has the potential to not only reduce chronic inflammation but also augment mitochondrial dysfunction and vaccine immunogenicity, whilst concomitantly providing additional antiviral and anti-inflammatory protection, afforded singularly by both components [11,23,28,31,50]. In addition to the protective aspects, host infection by SARS-CoV-2 acts to inhibit Beclin-1 and autophagosomal degradation, which can be reversed by increasing SPD [11,15,16,17].

Aside from the protective role of autophagy, dissemination of viral infection via phagocytosis is also vital, evidencing the importance of CD14 differentiated monocytes/macrophages by SPD and EUG described above. However, the viral removal is not limited to canonical autophagy. LC3-associated phagocytosis (LAP) is an alternative form of non-canonical autophagy, in which invading viruses are degraded in a single-membraned cargo-containing phagosome, or LAPosome [66,67,68]. 

Interestingly, aging cells deficient in LAP that fail to control pathogen infection are associated with increased inflammation and auto-immune disorders [66,67,68,69]. Although both canonical autophagy and LAP result in the lipidation of LC3-I to form LC3-II, LAP is controlled by some, but not all, members of the autophagy machinery [66,67]. Of relevance, LAP is stimulated by Rubicon, an autophagy-negative regulator, able to act on Beclin-1 [66,69,70]. This would imply antagonistic roles for SPD (stimulating autophagy) and Rubicon (stimulating LAP). 

Given that the regulatory mechanisms of LAP and autophagy are not well understood [70], it would be important to establish whether LAP induced phagocytosis is involved in removing SARS-CoV-2. Thereafter, it would be important to establish the effect of SPD and EUG on both LAP and canonical autophagy in the presence of viral infection. Testing in vitro the efficacy of a supplement containing SPD and EUG in stimulating autophagy and thereby also reducing inflammation, as well demonstrating SARS-CoV-2 anti-viral properties in numerous cell lines is the subject of current research.

## 4. Materials and Methods

### 4.1. Chemicals

The present study employed SPD obtained from natural source (wheat germ) instead of a synthetic reagent. A SPD extraction method was developed at the University of Bologna, associated with the Targeting Gut Disease (TGD) company (https://www.tgd.care/company) and a patent application made (number 102021000007331). In contrast to widely reported acid-based type extraction of SPD, a free-base extraction approach, followed by precipitation and filtration allowed to yield 10% SPD with reduced contents of undesirable components. 

The SPD extract was not in pure form. However, this did not detract from the efficiency of this compound. Given the smaller amounts of other polyamines in the natural extract, such as spermine, the SPD extract in the present experiment was initially compared to synthetic SPD to ensure comparable effects, which were evident (results not shown). Moreover, extracts were also analyzed previously to verify the absence of pesticides and pollutants possibly derived from the natural sources. The final product was suspended in ethanol. Pure EUG (>98%) obtained from clove bud essential oil, was provided from TGD and similarly diluted in ethanol with a purity of 99.5%.

Reagents for cell cultures, such as DMEM, RPMI, Fetal Bovine Serum (FBS), L-Glutamine, Penicillin-Streptomycin and rat tail collagen type I were purchased from GIBCO (Waltham, MA, USA). Bronchial Epithelial Cell Medium (BEpiCM) and Endothelial Cell Medium (ECM, including FBS and endothelial cell growth supplement) were obtained from ScienceCell (Carlsbad, CA, USA). MTT was from Life Technologies (Carlsbad, CA, USA). All other chemicals and solvents were of analytical grade.

### 4.2. Cell Model Systems and Growth Maintenance Conditions

L929 mouse fibroblasts (ATCC-CCL1) were cultured with DMEM, composed of 10% FBS, 1 mM L-glutamine and 1% penicillin-streptomycin. The Caco-2 human epithelial cell line (ATCC HTB-37), obtained from colorectal adenocarcinoma, was cultured with DMEM, supplemented with 10%, FBS and 1% penicillin-streptomycin. U937, a pro-monocytic, human myeloid leukemia cell line (ATCC CRL-1593.2), was cultured in RPMI-1640 medium, supplemented with 10% FBS and 1% penicillin-streptomycin. SHSY5Y neuronal cells (ATCC^®^ CRL2266) were cultured with DMEM, to which 15% FBS, 1 mM L-glutamine, 1% of non-essential amino acids and 1% penicillin-streptomycin, were added. 

Human bronchial epithelial cells (HBEpiC [ScienCell, catalogue number 3210]) were cultured in BEpiCM and human umbilical vein endothelial cells (HUVEC [ScienCell, catalogue number 8000]) in ECM. Stock cultures of all cell lines were maintained at 37 °C in a humidified atmosphere containing 5% CO_2_ in tissue culture flasks (75 cm^2^; BD Biosciences), and the culture medium changed every two days. Prior to experimentation, the Caco-2 cells were trypsinized and cell density evaluated microscopically using a Bürker counting chamber.

### 4.3. Experimental Monoculture Conditions for Autophagy Marker Detection and Cell Viability in Response to Spermidine and Eugenol Treatments

For the experimental analyses, L929 and Caco-2 cells (1 × 10^5^ cells/well) in complete medium were plated onto 96 well plates and incubated for 24 h. Thereafter, varying concentrations of SPD and EUG, both alone and in combination, were diluted in DMEM containing 10% BSA to determine the optimal concentrations without an effect on cell viability. U937 cells (in RPMI-1640 medium) were similarly exposed to varying SPD and EUG concentrations. 

The untreated control for each cell line contained only the culture medium and 10% Bovine Serum Albumin (BSA), whereas the starvation control contained complete medium without 10% BSA. Following either a 24 h exposure to the treatments, the medium was carefully aspirated, and the cell proliferation (MTT assay) and/or cell vitality (blue trypan exclusion assay) was measured.

Thereafter, the optimal substrate treatments were selected as follows: 0.3 mM SPD, 0.2 mM EUG and 0.3 mM SPD + 0.2 mM EUG. For the autophagy marker experiments, L929, Caco-2, U937, SHSY5Y, HBEpiC and HUVEC cells (1 × 10^5^ cells/well) in complete medium (DMEM, RPMI, BEpiCM or ECM) were, similarly, plated into four-well chamber slide plates. Thereafter, each cell line type was exposed for 4 h (as well as 24 h in preliminary experiments for the L929 and Caco-2 cell lines) to the three experimental substrate treatments. 

Controls (untreated control and starvation control) were also included. After the 4 h or 24 h exposure, cells were fixed and stained with a LC3-II marker and visualized with either immunofluorescent green (FG) or red chromogen. After either a 24 h exposure or 48 h exposure of all cell lines to the experimental treatments, viability was assessed using the MTT assay.

### 4.4. MTT and Blue Trypan Vitality Assays

After treatments (24 h or 48 h exposure periods), proliferative cells were detected using the MTT assay, according to the ISO 10993-5 International Standard procedure (ISO 10993-5, 2009). The MTT substrate was prepared in DMEM, added to cells in culture to attain a final concentration of 1 mg/mL and then incubated for 2 h at 37 °C with 5% CO_2_. After incubation, the medium was removed by aspiration. 

Isopropanol (100 µL) was added to each well, and formazan dye formation was evaluated by a multi-well scanning spectrophotometer at 540 nm. U937 suspension cells were plated at a concentration of 10^5^ cells/well onto 96-well culture plates in complete medium. After the treatments, the cells were incubated with 0.5% MTT solution in PBS for 4 h at 37 °C and then dissolved with 100 µL isopropanol in 0.04 N HCl. The plate was read at 540 nm, and the results were expressed as the percentage of viable cells with respect to untreated controls (70% ethanol). The percentage of cell proliferation was calculated using the following formula: absorbance value of treated sample/absorbance value of control × 100 = % of cell viability.

Cell viability was measured using the blue trypan exclusion assay trypan blue as reported in Truzzi et al. 2020 [33]. After the treatments (24 h exposure), cells were then carefully resuspended in a 0.4% Trypan Blue (Gibco) solution and vital cells were counted after 4 h using the Countess^®^II FL (Thermo Fisher Scientific, Waltham, MA, USA). The results were expressed as a viability percentage of the control.

### 4.5. Immunofluoresence and Immunocytochemistry for Autophagy LC3-II Marker Detection

Cell lines, plated and treated in the four-well Permanox chamber slides, were fixed in 70% ethanol for 10 min. Cells were permeabilized with 0.1% TritonX100 and stained for LC3-II (Novus) antibody and then labelled with Alexa Fluor 594-conjugated goat IgG (Thermo Fisher Scientific, Waltham, MA, USA) for fluorescence green imaging, according to the instructions provided by the manufacturer. Then slides were stained with 1 μg/mL propidium iodide (PI) solution, which stains the nuclear material red. Micrographs were taken on a Confocal Scanning Laser Microscopy (LeicaTCS4D; Leica, Exton, PA, USA).

For the immunocytochemical red chromogen staining, treated cell lines in the four-well permanox chamber slides were, similarly, fixed in 70% ethanol for 10 min. Cells were then permeabilized with 0,1% TritonX100 and stained for LC3-II (Novus) antibody and then labelled with UltraTek Alk-Phos Anti-Polyvalent (permanent red) Stain Kit (ScyTek Laboratories, Inc., Logan, UT, USA) according to the instructions provided. Quantification of staining was performed by analyzing six representative fields for each stained sample, using ImageJ software (Wayne Rasband, National Institute of Mental Health, Bethesda, MD, USA). The results were expressed as the percentage of LC3-II-positive cells.

U937 monocytes were incubated with anti-LC3-II antibody (Novus) for 20 min at 4 °C and then labelled with the secondary antibody Alexa Fluor 594-conjugated goat IgG for 20 min at 4 °C. Cells were analyzed using the Countess^®^II FL (Thermo Fisher Scientific, Waltham, MA, USA).

### 4.6. Western Blotting for LC3-II Expression

After the 24 h exposure of L929 and Caco-2 cells to the various treatments, the cell medium was washed with phosphate-buffered saline, PBS, and adherent cells were lysed on ice in Radio-Immunoprecipitation Assay buffer (RIPA buffer) pH 7.5. The total protein was quantified according to the Pierce™ BCA Protein Assay Kit (Thermo Fisher Scientific). Total protein (20 µg) was analyzed under reducing conditions on Bolt™ 12%, Bis-Tris Protein Gel (Thermo Fisher Scientific) and blotted onto nitrocellulose membranes. 

The blots were blocked for 1 min with SuperBlock™ T20 (PBS) Blocking Buffer (Thermo Fisher Scientific^®^) and incubated with anti-human LC3I-II (1:1000, Novus Biologicals) and beta-actin (1:1000, Invitrogen, Carlsbad, CA, USA) overnight at 4 °C. The membranes were then washed in PBS/Tween 20 and incubated with peroxidase-conjugated goat anti-mouse antibodies (1:10,000, Invitrogen) for 45 min at room temperature. Finally, the membranes were washed and developed using the Pierce™ Fast Western Blot Kit, ECL Substrate (Thermo Fisher Scientific). The band intensity was determined quantitatively using ImageJ software Fiji (Wayne Rasband) and LC3-II protein levels normalized to beta-actin expression.

### 4.7. Detection of Autophagy Protein Activation

The Human Autophagy Array C1 [AAH-ATG-1] (RayBio^®^ C-Series) for the semi-quantitative detection of 20 human proteins in cell and tissue lysates was purchased (RayBiotech, Norcross, GA, USA). Caco-2 cells (1 × 10^5^ cells/well) in complete medium were plated onto on four-well chamber slide plates and exposed for 4 h to cell medium (untreated control) and the three experimental substrate conditions (0.3 mM SPD, 0.2 mM EUG and 0.3 mM SPD + 0.2 mM EUG) and diluted in cell medium. 

Proteins were extracted from each of the treatments, and 200 µg were added to Antibody Array membranes and detected according to the manufacturer’s instructions. The expression of each of the individual 20 proteins for each treatment were then quantified using ImageJ software. According to the manufacturer’s instructions, the positive controls of the untreated control and three treatments were first normalized prior to the comparative quantification of the individual proteins.

### 4.8. Autophagy LC3-II Marker Detection in a Reconstituted Intestinal Cell Model with Caco-2, U937 and L929 Cells

3D reconstituted intestinal equivalents were constructed using Caco-2, U937 and L929 cells. The 3D cultures were performed as described previously [33]. Cell-free 0.5 mg/mL collagen solution (1.35 mg/mL rat tail collagen type I in DMEM with 10% fetal bovine serum, FBS and 1% Pen/Strep) was added to tissue culture inserts (Transwell, Costar, Cambridge, MA, USA) in 12-well plates. This pre-coated layer was overlaid with 1 mL of L929 fibroblasts (1 × 10^5^/mL) together with U937 monocytes (3 × 10^4^/mL) mixed with collagen type I. 

After incubation (2 h) at 37 °C, Caco-2 cells (2 × 10^5^) were seeded onto dermal reconstructs and incubated at 37 °C with Caco-2 medium, added both on the upper and the lower part of the filter support. After five days, the cell models were treated with the treatments (0.3 mM SPD, 0.2 mM EUG and 0.3 mM SPD + 0.2 mM EUG) or DMEM only for the CNTL for 24 h. For the immunohistochemical analyses, the cells were fixed with formalin for 2 h at room temperature, dehydrated and embedded in paraffin. 

Paraffin-embedded reconstituted 3D intestinal equivalents were then rehydrated, and sections (4 μm thick) were stained with hematoxylin and eosin (H&E). For the CD14 staining, sections were stained with CD14 (GeneTex Inc). Immunohistochemistry was performed using fast red chromogen according to the UltraTek Alk-Phos Anti-Polyvalent (permanent red) Stain Kit (ScyTek Laboratories, Inc.). For the LC3-II staining, the cells were permeabilized with 0.1% TritonX100 and stained for LC3-II (Novus) antibody and then labelled with UltraTek Alk-Phos Anti-Polyvalent (permanent red) Stain Kit, as described previously.

### 4.9. Autophagy LC3II Marker Detection and Viability of Caco-2 and L929 Cells in the Presence of the Inflammation Elicitor Lipopolysaccharide

Monoculture experiments were performed using cell lines plated into 96-well tissue culture plates (10^5^ cells/well) in complete medium. After 24 h, cells were pretreated with each of the treatments (0.1 mM SPD, 0.05 mM EUG and 0.1 mM SPD + 0.05 mM EUG) in complete medium for 1 h. Thereafter, LPS (1 ng/mL) was added to each of the treatments and the cells incubated for 24 h. The untreated control contained the same volume of cell medium without LPS or the treatments. Cells treated with LPS alone (1 ng/mL in DMEM) without the pre-treatments were also included. After 24 h, cells were tested for viability using the MTT assay or harvested for Western Blotting analysis.

### 4.10. Statistical Analysis

Cell tests were performed using multiple replicates. Data are expressed in the form of bar graphs showing the mean for each treatment as well as the positioning of the individual replicates within each treatment bar. Statistical analysis was conducted using GraphPad Prism Version 9.3.1 (2021). Significance was determined by one-way variance (ANOVA) and the Turkey–Kramer test to any significant differences between treatments at *p* < 0.01.

## 5. Conclusions

The present study verified the stimulation of autophagy by SPD, corroborating previous research. Moreover, we addressed the lack of information pertaining to the in vitro effects of EUG on autophagy in healthy cell lines. As with SPD, EUG was shown to induce autophagy without cytotoxic effects in the five non-cancerous cell lines (HBEpiC, SHSY5Y, HUVEC, L929 and U937). In the Caco-2 cells, there was a cytotoxic effect, although the effect was minimal. Increased autophagy was demonstrated through LC3-II accumulation in all six cell lines as well as through the upregulation of several core autophagy proteins in Caco-2 cells, denoting complete autophagic flux. 

Verification of complete flux was also implicated from the activation and differentiation of U937 monocytes within the reconstituted intestinal model. Of interest, the unique combination treatment (SPD + EUG) yielded comparable increases, compared to SPD and EUG administered alone. In Caco-2 and L929 cells, increased autophagy reduced LPS-induced inflammatory effects on cell proliferation. By stimulating autophagy and thereby reducing inflammation, a supplement comprised of both SPD and EUG could afford a protective effect against SARS-CoV-2 in combination with vaccination programs. Moreover, such a supplement could also be used in the treatment of COVID-19 to prevent more severe outcomes.

## Figures and Tables

**Figure 1 molecules-27-03425-f001:**
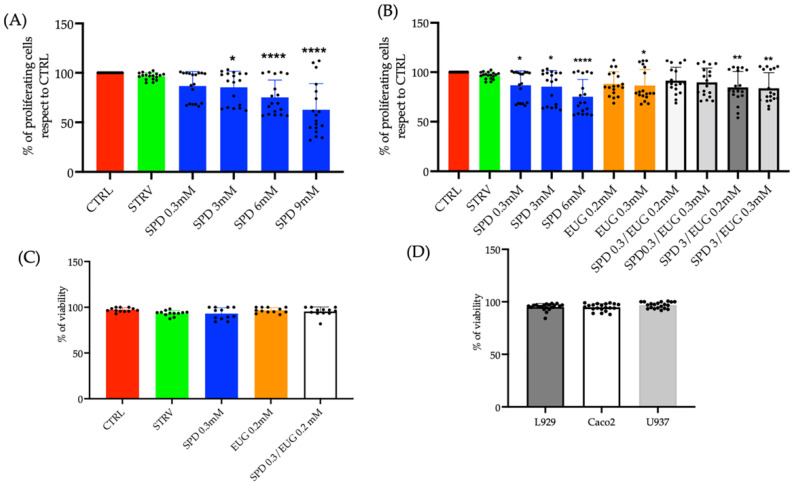
(**A**) Effect of varying spermidine (SPD) concentrations on cellular proliferation in Caco-2, L929 and U939 lines after 24 h, reported as an average response on all three cell lines using the MTT assay. (**B**) Effect of eugenol (EUG) concentration in combination with SPD on Caco-2, L929 and U937 lines after 24 h, reported as an average response using the MTT assay. Comparison was made to the untreated control and starvation-induced cells (STARV). (**C**) The effect on cell viability using the blue trypan exclusion assay after a 24 h exposure to 0.3 mM SPD and 0.2 mM EUG alone and in combination (0.3 mM SPD + 0.2 mM EUG) on Caco-2, L929 and U939 lines expressed as an average response or (**D**) as a response of the individual cell lines. The designation of the number of stars *, ** and **** represents significant differences between treatments as determined by one-way ANOVA at the 99% confidence level (*p* < 0.01). The black dots indicate the positioning of the individual replicates within the bar for each sample.

**Figure 2 molecules-27-03425-f002:**
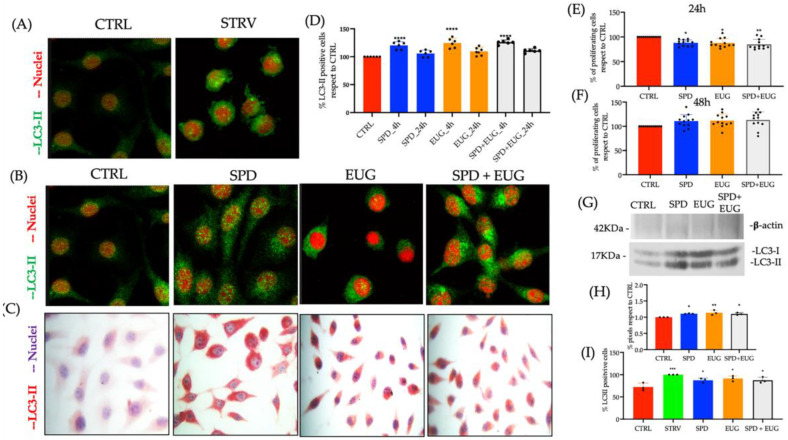
The effect of 0.3 mM spermidine (SPD) and 0.2 mM eugenol (EUG), alone and in combination (0.3 mM SP + 0.2 mM EUG) on LC3-II activation and cell viability in L929 (**A**–**H**) and U937 (**I**) cell lines. (**A**) LC3-II in the untreated control and the starvation (STARV)-induced control was detected with immunofluorescence green staining (FG) after 4 h. (**B**,**C**) For the SPD, EUG and SPD + EUG treatments, LC3-II was detected with FG staining and red chromogen staining. (**D**) Statistical analysis of the LC3-II positive cells stained with red chromogen was performed after both 4 h and 24 h from the start of all treatments. Viability (MTT assay) of the cells after a (**E**) 24 h exposure and a (**F**) 48 h exposure to the treatments. (**G**) Western blotting of LC3-II with respect to β-actin was performed on L929 cells that were untreated or treated with SPD, EUG and SPD + EUG and the pixels quantified (H). (I) Quantification of LC3-II positive cells of U937 stained with FG after 4 h following starvation, SPD, EUG and SPD + EUG treatments. The number of stars *, **, *** and **** represents significant differences between treatments as determined by one-way ANOVA at the 99% confidence level (*p* < 0.01). The black dots indicate the positioning of the individual replicates within the bar for each sample.

**Figure 3 molecules-27-03425-f003:**
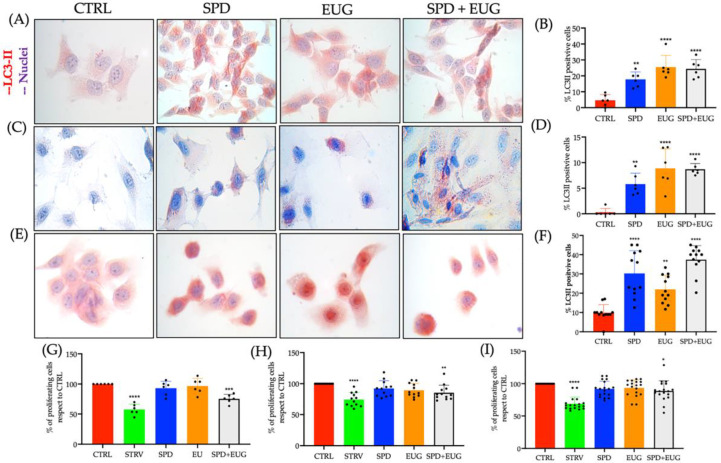
The effect of 0.3 mM spermidine (SPD) and 0.2 mM eugenol (EUG), alone or in combination (0.3 mM SP + 0.2 mM EUG) on LC3-II activation in SHSY5Y (**A**,**B**), HUVEC (**C**,**D**) and HBEpiC (**E**,**F**) cell lines after 4 h. LC3-II was detected by fast red chromogen staining (**A**,**C**,**E**) and the percentage of LC3-II positive cells reported compared to untreated control cells (**B**,**D**,**F**). The MTT assay for all treatments was performed and the data reported as percentage of the untreated control for (**G**) SHSY5Y (**H**), HUVEC and (**I**) HBEpiC cells. The number of stars *, **, *** and **** represents significant differences between treatments as determined by one-way ANOVA at the 99% confidence level (*p* < 0.01). The black dots indicate the positioning of the individual replicates within the bar for each sample.

**Figure 4 molecules-27-03425-f004:**
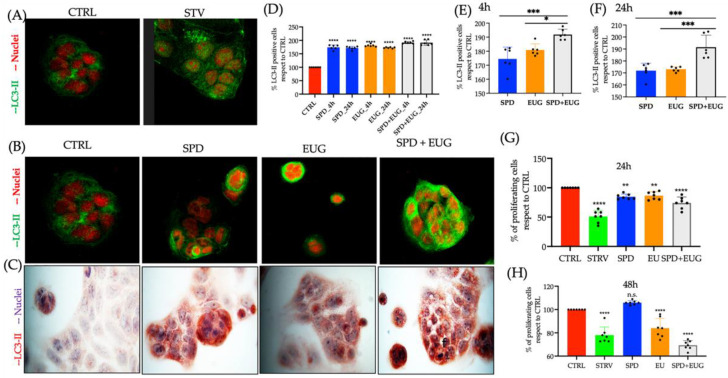
The effect of 0.3 mM spermidine (SPD) and 0.2 mM eugenol (EUG), alone and in combination (0.3 mM SP + 0.2 mM EUG) on LC3-II activation and viability in Caco-2 cell lines. (**A**) LC3-II staining in the untreated control cells), and the starvation (STARV)-induced control cells was detected with immunofluorescence green staining (FG) after 4 h. (**B**,**C**) For the SPD, EUG and SPD + EUG treatments, LC3-II was detected with FG staining and fast red chromogen staining. (**D**–**F**) Statistical analysis of the LC3-II positive cells stained with red chromogen was performed after both 4 h and 24 h from the start of all treatments. MTT assay performed after 24 h (**G**) and 48 h (**H**) on Caco-2 cells that were untreated or treated with SPD, EUG and SPD + EUG. The number of stars *, **, *** and **** represents significant differences between treatments as determined by one-way ANOVA at the 99% confidence level (*p* < 0.01), whereas n.s. is not significant.

**Figure 5 molecules-27-03425-f005:**
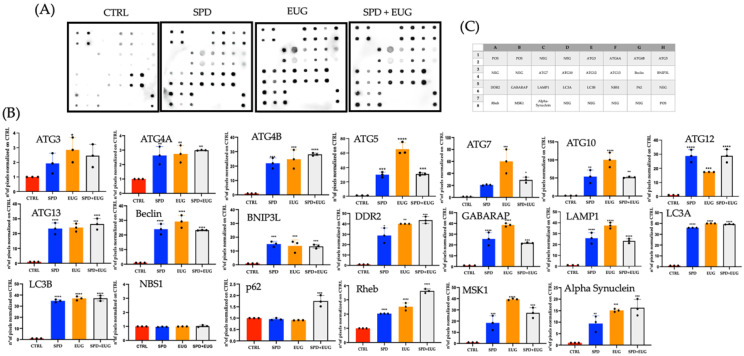
Expression (**A**) and expression quantification (**B**) of 20 individual human autophagy proteins, identified from the position location on the micro-array (**C**). Expression of the proteins (**A**) in the control (CTRL), 0.3 mM spermidine (SPD), 0.2 mM eugenol (EUG), 0.3 mM SP + 0.2 mM EUG (SPD + EUG) treated cells. Protein expression for each of the 20 proteins was quantified (**B**) for each treatment. The number of stars *, **, *** and **** represents significant differences between treatments as determined by one-way ANOVA at the 99% confidence level (*p* < 0.01). The three black dots indicate the positioning of the individual replicates within the bar for each sample.

**Figure 6 molecules-27-03425-f006:**
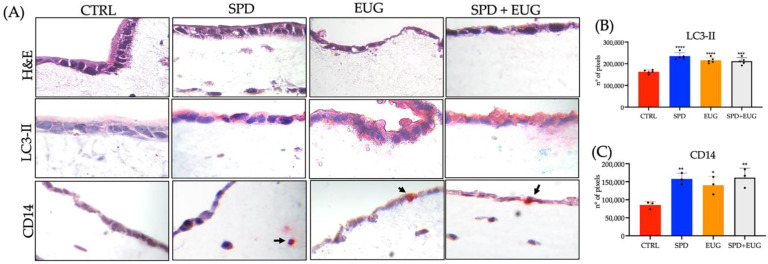
(**A**) Hematoxylin and eosin (H&E), LC3-II staining and CD14 staining of U937 monocytes in 3D intestinal equivalents (Caco-2 /U937/L929 co-cultures) exposed to 0.3 mM spermidine (SPD) and 0.2 mM eugenol (EUG) and in combination (0.3 mM SP + 0.2 mM EUG) compared to the untreated control. Arrows indicate the presence of monocytes (CD14 positive cells). The magnification was ×40. (**B**) Quantification of the LC3-II stained Caco-2 cells and (**C**) CD14 stained U937 cells. The number of stars *, **, *** and **** represents significant differences between treatments as determined by one-way ANOVA at the 99% confidence level (*p* < 0.01). The black dots indicate the positioning of the individual replicates within the bar for each sample.

**Figure 7 molecules-27-03425-f007:**
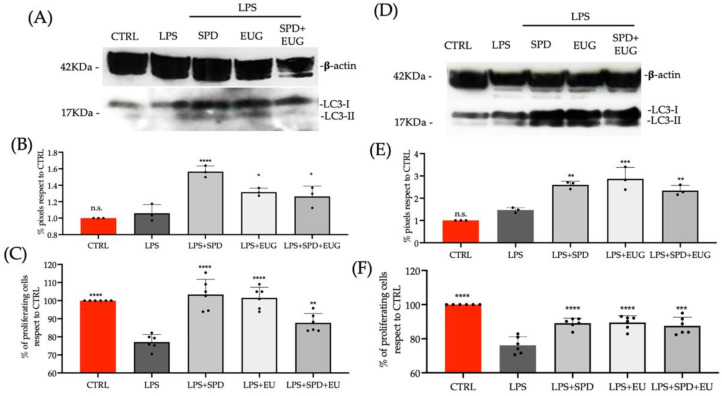
The effect of a 1 h pretreatment with 0.3 mM spermidine (SPD) and 0.2 mM eugenol (EUG), alone and in combination (0.1 mM SPD + 0.05 mM EUG), before the administration of lipopolysaccharide (LPS, 1ng/mL) to Caco-2 cells (**A**–**C**) and L929 cells (**D**–**F**) for 24 h (**A**,**D**). Treatments were compared with the untreated control and LPS treatment alone. Western blotting of LC3-II with respect to β-actin was performed on Caco-2 cells (**A**) and L929 cells (**D**) that were untreated or treated with LPS + SPD, LPS + EUG and LPS + SPD + EUG and the pixels quantified (**B**,**E**). Cell viability using the MTT assay on the Caco-2 (**C**) and L929 (**F**) cell lines. The number of stars *, **, *** and **** represents significant differences between treatments as determined by one-way ANOVA at the 99% confidence level (*p* < 0.01), whereas n.s. is not significant. The black dots indicate the positioning of the individual replicates within the bar for each sample.

## Data Availability

Not applicable.

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
