# Peer review of "Wheat Germ Spermidine and Clove Eugenol in Combination Stimulate Autophagy In Vitro Showing Potential in Supporting the Immune System against Viral Infections"

_molecules, 2022, doi:10.3390/molecules27113425_

Round 1

Reviewer 1 Report

The authors prepared a manuscript aiming to report the ability of wheat germ spermidine and clove eugenol (in combination) to stimulate autophagy and to reduce inflammation, which might be potential as one supplementary to be used in the COVID-19 treatment. In the present manuscript, the authors demonstrated, using an in vitro approach, that the combination of spermidine and eugenol can induce the LC3-dependent autophagy.

The manuscript is well-written and well-prepared. I found that the topic of current MS is adequate and acceptable for the journal’s scope. The story is compelling and there is a novelty. It is worthy of publication. However, some concerns/questions, as mentioned below, need to be addressed by the authors before being considered for publication in this journal.

First, while the authors have argued for the importance of this topic, the novelty of this manuscript remains overlooked. The introduction section written in the manuscript is quite short to provide an insight into why this manuscript is important. Especially with the unclear reasons on why the combination of compounds needs to be done. I am wondering why the authors need to combine the spermidine and eugenol? Why choose these two compounds but not other compounds that have been reported to be potential. Let’s say for example EGCG from green tea? I recommend the authors extend the introduction section to emphasize the importance of this manuscript and the reasons why the authors perform a combination.

Second, as the authors presented in their manuscript, in most of the bar graphs, the authors used “% respect to the control” on the Y-axis. For example, Graph 1A-B, 2D-F, and 2H, some of Fig 3, etc. Using this approach, the authors did not show any SD. I am strongly recommending the authors used another approach to visualize the data with SD value so that readers can see how many repetitions have been done in the experiments and the reproducibility has been achieved properly.

Third, the authors' data provide an indication that autophagy might serve as an important factor in the protection of the infected host/cells against SARS-CoV-2. Indeed, several literatures have suggested that autophagy provides antiviral protection. However, other literatures also point out the role of autophagy in the dissemination of viral infection. Owing to the fact that autophagy has a double-edge function in the event of viral infection, we cannot consciously diminish the role of other immune effectors. For example, LC3-mediated phagocytosis. This shall be discussed properly to provide insights into the possibilities.

Fourth, the use of Covid-19 in the title is misleading. I highly recommend not to use it. Even though it is fine that the authors suggest the potential of an autophagy-related approach for dealing with Covid-19, the authors need to be careful in suggesting that the Covid-19 risk groups can get benefit from this treatment without any evidence in the experiments carried out by the authors.

Finally, I recommend that this study shall be considered for publication in this journal after addressing the comments/concerns above.

Author Response

Comments and Suggestions for Authors

First, while the authors have argued for the importance of this topic, the novelty of this manuscript remains overlooked. The introduction section written in the manuscript is quite short to provide an insight into why this manuscript is important. Especially with the unclear reasons on why the combination of compounds needs to be done. I am wondering why the authors need to combine the spermidine and eugenol? Why choose these two compounds but not other compounds that have been reported to be potential. Let’s say for example EGCG from green tea? I recommend the authors extend the introduction section to emphasize the importance of this manuscript and the reasons why the authors perform a combination.

The reviewer is correct. In the Introduction, we have inserted the answer to this question (in blue) that follows the paragraph already present (in black):

“To this end, the present study, based on the results of Gassen et al. [11], aimed to investigate the effect of wheat germ SPD extract, in a unique combination with eugenol (EUG) on autophagy. This incentive formed part of one of the European Institute for Innovation & Technology (EIT) FOOD (www.eitfood.eu) projects using natural products that can support either the prevention of COVID-19 or in the treatment of individuals with a higher risk for more severe outcomes of COVID-19. Although, COVID-19 severity is well-known to vary from asymptomatic to mild symptoms (such as fever, cough, shortness of breath, sore throat, muscle ache and gastrointestinal symptoms), in some individuals severe disease progression has been linked to the so-called “cytokine storms”, initiated through rapid virus propagation and uncontrolled inflammation. These individuals include the elderly and those with well-known comorbidities related to systemic inflammation, such as obesity, atherosclerosis, type 2 diabetes, hypertension, and asthma [28]. EUG has long been of interest as an anti-viral compound, effective at inhibiting the replication process of a numerous viruses [29-30], and of recent potential interest in the treatment of SARS-CoV-2 [31].   

The basis for selecting a dual therapy was the potential advantage of using two components with complementary attributes such as anti-viral and anti-inflammatory activities [28]. SPD was selected for reported benefits of increasing autophagy and inhibiting SARS-CoV-2 propagation [11]. Instead, EUG was selected for known anti-viral activity [29-31]. In addition, there is a particular interest in our research group for the anti-oxidant and anti-inflammatory properties of EUG-rich essential oils [32]. Interestingly, essential oil single molecules, such as EUG, are natural and inexpensive, and as such are lacking sponsors for trials needed to validate their therapeutic efficacy [32]. However, a low production cost would render such a potential component more accessible to the public and for this reason EUG was selected for investigation in combination with SPD. Unlike SPD which is reported to stimulate autophagy, from a single paper using cell lines, EUG was only proposed as an inhibitor of autophagy [33]. Therefore, it was important to test whether EUG did not produce an antagonistic effect with SPD on autophagy. Hence, the objective of the present study was to investigate the autophagy efficacy of SPD and the potential effect on autophagy exerted by EUG, both alone and in combination, respectively. To this end a series of human cell lines were used, including those from lung, brain, and intestinal origins, organs most affected by COVID-19. The SPD and EUG were derived from natural plant sources, as the objective was to simulate natural nutritional intake of the active components. In order to be administered in small doses, the active ingredients were used on cells in a concentrated form (SPD and pure EUG).

An additional reference was inserted as new number 32

Spisni, E.; Petrocelli, G.; Imbesi, V.; Spigarelli, R.; Azzinnari, D.; Donati Sarti, M.; Campieri, M.; Valerii, M.C. Antioxidant, anti-inflammatory, and microbial-modulating activities of essential oils: Implications in colonic pathophysiology. Int J Mol Sci. 2020, 21(11), 4152. https://doi.org/10.3390/ijms21114152.

Second, as the authors presented in their manuscript, in most of the bar graphs, the authors used “% respect to the control” on the Y-axis. For example, Graph 1A-B, 2D-F, and 2H, some of Fig 3, etc. Using this approach, the authors did not show any SD. I am strongly recommending the authors used another approach to visualize the data with SD value so that readers can see how many repetitions have been done in the experiments and the reproducibility has been achieved properly.

An alternative approach was used that indicates the exact positioning of individual replicates within each bar. All the figures have been changed and now show the readers how many repetitions were made in the experiments and that the reproducibility was achieved properly.

This can be illustrated from Figure 3 as an example

In the methods section the following has been included: “Cell tests were carried out using multiple replicates. Data was expressed in the form of bar graphs showing the mean for each treatment as well as the positioning of the individual replicates within each treatment bar, respectively” (line 709).

Third, the authors' data provide an indication that autophagy might serve as an important factor in the protection of the infected host/cells against SARS-CoV-2. Indeed, several literatures have suggested that autophagy provides antiviral protection. However, other literatures also point out the role of autophagy in the dissemination of viral infection. Owing to the fact that autophagy has a double-edge function in the event of viral infection, we cannot consciously diminish the role of other immune effectors. For example, LC3-mediated phagocytosis. This shall be discussed properly to provide insights into the possibilities.

This aspect has been included in the discussion. The following has been added after the sentence below:

By stimulating autophagy, the contribution of both SPD and EUG has the potential, not only to reduce chronic inflammation but also augment mitochondrial dysfunction and vaccine immunogenicity, whilst concomitantly providing additional antiviral and anti-inflammatory protection, afforded singularly by both components [11,23,28,31,50].

In addition to the protective aspects, host infection by SARS-CoV-2 acts to inhibit Beclin-1 and autophagosomal degradation, which can be reversed by increasing SPD [11,15-17]. Aside from the protective role of autophagy, dissemination of viral infection via phagocytosis is also vital, evidencing the importance of CD14 differentiated mono-cytes/macrophages by SPD and EUG described above. However, the viral removal is not limited to canonical autophagy. LC3-associated phagocytosis (LAP) is an alternative form of non-canonical autophagy, in which invading viruses are degraded in a single-membraned cargo-containing phagosome, or LAPosome [67-68]. Interestingly, aging cells deficient in LAP that fail to control pathogen infection are associated with increased inflammation and auto-immune disorders [67-69]. Although both canonical autophagy and LAP result in the lipidation of LC3-I to form LC3-II, LAP is controlled by some, but not all, members of the autophagy machinery [67-68]. Of relevance, LAP is stimulated by Rubicon, an autophagy-negative regulator, able to act on Beclin-1 [67,70-71]. This would imply antagonistic roles for SPD (stimulating autophagy) and Rubicon (stimulating LAP). Given that the regulatory mechanisms of LAP and autophagy are not well understood [71], it would be important to establish whether LAP-induced phagocytosis is required in removing SARS-CoV-2. Thereafter, it would be important to establish the effect of SPD and EUG on both LAP and canonical autophagy in the presence of viral infection.

Five additional references were added (reference 66-70) which have been included in the Reference section.

  1. Martinez, J. LAP it up, fuzz ball: a short history of LC3-associated phagocytosis. Curr Opin Immunol. 2018, 55, 54-61. https://doi.org/10.1016/j.coi.2018.09.011.
  2. Martinez, J. Detection of LC3-associated phagocytosis (LAP). Curr Protoc Cell Biol. 2020, 87, e104. https://doi.org/10.1002/cpcb.104.
  3. Inomata, M.; Xu, S.; Chandra, P.; Meydani, S.N.; Takemura, G.; Philips, J.A.; Leong, J.M. Macrophage LC3-associated phagocytosis is an immune defense against Streptococcus pneumoniae that diminishes with host aging. Proc Natl Acad Sci USA . 2020, 117(52), 33561-33569. https://doi.org/10.1073/pnas.2015368117.
  4. Matsunaga, K.; Saitoh, T.; Tabata, K.; Omori, H.; Satoh, T.; Kurotori, N.; et al. Two Beclin 1-binding proteins, Atg14L and Rubicon, reciprocally regulate autophagy at different stages. Nat. Cell Biol. 2009, 11, 385–396. https://doi.org/10.1038/ncb1846.
  5. Yamamoto-Imoto, H.; Minami, S.; Shioda, T.; Yamashita, Y.; Sakai, S.; Maeda, S.; Yamamoto, T.; Oki, S.; Takashima, M.; Yamamuro, T. et al. Age-associated decline of MondoA drives cellular senescence through impaired autophagy and mitochondrial homeostasis. Cell Rep. 2022, 38, 110444. https://doi.org/10.1016/j.celrep.2022.110444.

Fourth, the use of Covid-19 in the title is misleading. I highly recommend not to use it. Even though it is fine that the authors suggest the potential of an autophagy-related approach for dealing with Covid-19, the authors need to be careful in suggesting that the Covid-19 risk groups can get benefit from this treatment without any evidence in the experiments carried out by the authors.

We have changed the title to read:

Wheat germ spermidine and clove eugenol in combination stimulate autophagy in vitro showing potential in supporting the immune system against viral infections

Finally, I recommend that this study shall be considered for publication in this journal after addressing the comments/concerns above.

Reviewer 2 Report

The manuscript shows the effect of spermidine and eugenol, both alone and in combination, on autophagy in different cell lines showing a significant enhancement of autophagy without significant cytotoxic effects.

Remarks:

Why the substances were isolated from a natural sources? It may always contain some contaminations.

Title: Would spermidine not from wheat and eugenol not from clove be not efficient?

Line 58: “has been shown to utilizes”, please change to “has been shown to utilize”

Line 126: please check the correctness of the name of the compound

Lines 134/135: “incubation from 3 to 9 13 mM SPD”, please correct the sentence

Lines 138/139: “0.2 mM SPD and 0.3 mM EUG alone, or the combined treatment of 0.3 mM SPD and 0.2 mM EUG”, while different concentrations for the combined treatment? It would be illogical; apparently a numerical error, please correct

Figure 1B. 0.3 mM SPD shows some negative effect of 0.3 mM SPD, in opposition to what is stated in the text (Line 140). What is the quantity on the axis of ordinates? “% respect…” is the unit, not the quantity. The same remark concerns Fig. 2E-I, Fig. 3G-I, Fig. 4G,H, Fig. 6B,C and Fig. 7B-F.

Fig. 5: Is it possible to quantify the effects?

Line 690 should be deleted

Author Response

Comments and Suggestions for Authors

Remarks:

  1. Why the substances were isolated from a natural sources? It may always contain some contaminations.

The first part of the question has been included in the aim of the introduction as follows:

The SPD and EUG were derived from natural plant sources, as the objective was to simulate natural nutritional intake of the active components. In order to in order to be administered in small doses, the active ingredients were in a concentrated form (SPD and pure EUG).

The second part of the question has been included in the methods and materials describing the extraction. See below:

The SPD extract was not in pure form. However, this did not detract from the efficiency of this compound. Given the smaller amounts of other polyamines in the natural extract, such as spermine, the SPD extract in the present experiment was initially compared to synthetic SPD to ensure comparable effects, which were evident (results not shown). Moreover, extracts were also analyzed previously to verify the absence of pesticides and pollutants possibly derived from the natural sources. The final product was suspended in ethanol. Pure EUG (>98%) obtained from clove bud essential oil, was provided from TGD, and similarly diluted in ethanol with a purity of 99.5%.

  1. Title: Would spermidine not from wheat and eugenol not from clove be not efficient?

We wish to indicate the natural sources from which SPD and EUG were obtained. The reason is that although SPD and EUG were the principle ingredients of wheat germ and clove oil, respectively, SPD was not in pure form within the extracts that was used on the cells. Instead EUG should be considered pure, even if obtained from clove bud essential oil.

  1. Line 58: “has been shown to utilizes”, please change to “has been shown to utilize”

This has been corrected

  1. Line 126: please check the correctness of the name of the compound.

The name of the compound 3-(4,5-dimetiltiazol-2-il)-2,5-difeniltetrazolio (MTT) was incorrect and has been changed to the correct name in English as defined by Sigma Aldrich and now reads as:

3-(4,5-Dimethyl-2-thiazolyl)-2,5-diphenyl-2H-tetrazolium bromide

  1. Lines 134/135: “incubation from 3 to 9 13 mM SPD”, please correct the sentence.

The sentence now reads: In contrast, at concentrations ranging from 3 to 9 mM, SPD was shown to significantly reduce proliferation

  1. Lines 138/139: “0.2 mM SPD and 0.3 mM EUG alone, or the combined treatment of 0.3 mM SPD and 0.2 mM EUG”, while different concentrations for the combined treatment? It would be illogical; apparently a numerical error, please correct.

Yes it was a numerical error. The sentence now reads:

to either the starvation control, 0.3 mM SPD and 0.2 mM EUG alone, or the combined treatment (0.3 mM SPD and 0.2 mM EUG) on the three cell lines

  1. Figure 1B. 0.3 mM SPD shows some negative effect of 0.3 mM SPD, in opposition to what is stated in the text (Line 140). What is the quantity on the axis of ordinates? “% respect…” is the unit, not the quantity. The same remark concerns Fig. 2E-I, Fig. 3G-I, Fig. 4G,H, Fig. 6B,C and Fig. 7B-F.  There was a slight negative effect after 24 h and this has been changed in the article to acknowledge this. After 4 h, the time required to stimulate autophagy, there was no negative effect on the cells using the MTT assay. Noteworthy, when using the Blue trypan viability test, there was no negative effect on the Caco-2, L929 or U937 cells after 24 h (Figure 1C and Figure 1D).

The Y axis respecting the percentages have been expressed to indicate what they are based on. We apologize for the vague axes   

  1. Fig 5: Is it possible to quantify the effects?

In the methods section under the micro-array this following was included:

The expression of each of the individual 20 proteins for each treatment were then quantified using ImageJ software. According to the manufacturer’s instructions, the positive controls of the untreated control and three treatments were first normalized prior to the comparative quantification of the individual proteins.

Figure 5 has been modified as well as the figure legend as follows:

Figure 5. Exposure of Caco-2 cells (A) to an array of 20 human autophagy proteins (C) for 4 hours. Expression of the proteins (A) in the control (CTRL), 0.3 mM spermidine (SPD), 0.2 mM eugenol (EUG), 0.3 mM SP + 0.2 mM EUG (SPD + EUG) treated cells. Protein expression for each of the 20 proteins was quantified (B) for each treatment. The number of stars represents significant differences between treatments as determined by one-way ANOVA at the 99% confidence level (p < 0.01). The three black dots indicate the positioning of the individual replicates within the bar for each sample.

Then where applicable modifications have been made in the text of the results.

  1. Line 690 should be deleted

Line 690 has now been deleted

Round 2

Reviewer 1 Report

The authors have revised the manuscript in a proper manner. I found that the manuscript now is adequate and acceptable. After the revision, the manuscript is clearly very well-prepared and has been written at a proper academic level to meet the high standards of the Molecules. The authors also have addressed my comments in an appropriate manner. To this end, I recommend the manuscript shall now be considered for publication in the journal in its current form. This manuscript shall serve as a great addition to the scientific literature.

Congratulations.